# Antimicrobial Susceptibility Testing: A Comprehensive Review of Currently Used Methods

**DOI:** 10.3390/antibiotics11040427

**Published:** 2022-03-23

**Authors:** Ina Gajic, Jovana Kabic, Dusan Kekic, Milos Jovicevic, Marina Milenkovic, Dragana Mitic Culafic, Anika Trudic, Lazar Ranin, Natasa Opavski

**Affiliations:** 1Institute of Microbiology and Immunology, Faculty of Medicine, University of Belgrade, 11000 Belgrade, Serbia; jovana.kabic@gmail.com (J.K.); dusan_vk@yahoo.com (D.K.); milos.jovicevic@med.bg.ac.rs (M.J.); lazarranin58@gmail.com (L.R.); natasaopavski@gmail.com (N.O.); 2Department of Microbiology and Immunology, Faculty of Pharmacy, University of Belgrade, 11000 Belgrade, Serbia; marina.milenkovic@pharmacy.bg.ac.rs; 3Faculty of Biology, University of Belgrade, 11000 Belgrade, Serbia; mdragana@bio.bg.ac.rs; 4Faculty of Medicine, University of Novi Sad, 21000 Novi Sad, Serbia; anika.trudic@mf.uns.ac.rs; 5Institute for Pulmonary Diseases of Vojvodina, Sremska Kamenica, 21204 Novi Sad, Serbia

**Keywords:** antimicrobial susceptibility testing, antimicrobial resistance, methods

## Abstract

Antimicrobial resistance (AMR) has emerged as a major threat to public health globally. Accurate and rapid detection of resistance to antimicrobial drugs, and subsequent appropriate antimicrobial treatment, combined with antimicrobial stewardship, are essential for controlling the emergence and spread of AMR. This article reviews common antimicrobial susceptibility testing (AST) methods and relevant issues concerning the advantages and disadvantages of each method. Although accurate, classic technologies used in clinical microbiology to profile antimicrobial susceptibility are time-consuming and relatively expensive. As a result, physicians often prescribe empirical antimicrobial therapies and broad-spectrum antibiotics. Although recently developed AST systems have shown advantages over traditional methods in terms of testing speed and the potential for providing a deeper insight into resistance mechanisms, extensive validation is required to translate these methodologies to clinical practice. With a continuous increase in antimicrobial resistance, additional efforts are needed to develop innovative, rapid, accurate, and portable diagnostic tools for AST. The wide implementation of novel devices would enable the identification of the optimal treatment approaches and the surveillance of antibiotic resistance in health, agriculture, and the environment, allowing monitoring and better tackling the emergence of AMR.

## 1. The Emergence of Antimicrobial Resistance and Overlooked Pandemic

Antimicrobial resistance (AMR) remains the world’s most urgent public health concern [1]. According to the World Health Organization (WHO), Geneva, Switzerland, antibiotic resistance is rising to dangerously high levels in all parts of the world, leading to increased morbidity and mortality [2]. Hence, the six leading mortality-causing pathogens—*Escherichia coli*, *Staphylococcus aureus*, *Klebsiella pneumoniae*, *Streptococcus pneumoniae*, *Acinetobacter baumannii*, and *Pseudomonas aeruginosa*—were responsible for 929,000 deaths attributable to AMR and 3.57 million deaths associated with AMR in 2019 [3]. This number could rise to 10 million by 2050 according to estimates by the WHO [4]. Furthermore, the SARS-CoV-2 pandemic has exacerbated the existing global crisis of AMR, mostly due to the mis- and over-use of antibiotics, treatments that induce immunosuppression, and prolonged hospitalisation [5]. Besides, during the COVID-19 pandemic, limited ability to work with AMR partnerships, decreases in funding, and reduced availability of nursing, medical, and public health staff affected AMR surveillance, prevention, and control [6]. In addition, increased use of disinfectants, including hand sanitisers and surface cleaners, is anticipated to cause increased rates of antimicrobial resistance in pathogenic microbes in the coming years [7]. Replacement of first-line antibiotics by more expensive medications, a longer duration of illness, and treatment-related to AMR increases healthcare costs as well as the economic burden on patients and societies [2]. The World Bank estimates that drug-resistant infections could cause a global economic crisis, leading to 28 million people who could be pushed into extreme poverty every year by 2050, with an overall cost to the global economy of USD 1 trillion per year [8].

Throughout their evolution, bacteria have developed versatile resistance mechanisms to antibiotics. The four main mechanisms of AMR are enzymatic inactivation of antimicrobial compounds, alteration of a drug target, reduced permeability of the outer membrane, and active drug efflux [9]. Hydrolases (e.g., beta-lactamases encoding by *bla* genes, such as extended-spectrum beta-lactamases, ESBL; cephalosporinases; and carbapenemases), passivation, and modified enzymes are three of the most important drug-inactivating enzymes. An altered target site is a major cause of Gram-positive bacteria’s drug resistance (e.g., PBP2a in methicillin-resistant *S. aureus*, MRSA by the acquisition of the *mecA* gene and other homologues), as well as polymyxin-resistant bacteria. The membrane permeability is a key in the level of susceptibility to antibiotics in some bacteria, such as *Enterobacterales*. Modification of the bacterial envelope by decreasing the porin production or increasing the expression of efflux pump systems (e.g., M phenotype in *Streptococcus* spp. encoding by *mefA* gene) has been reported [10]. 

The causes of antimicrobial resistance are complex and multifaceted. In countries where antibiotics are sold without a prescription or used as growth-promoting substances or prophylactic additives in livestock farming, antibiotic-resistant bacteria develop especially fast [2]. Administration of antibiotics to patients with suspected moderate to severe bacterial infections has been deemed inappropriate in at least half of the cases [11]. Antimicrobial stewardship (AMS) is one of the key strategies for combatting resistance. Implementation of such programs is therefore recommended across the globe [12]. 

The present review provides an updated overview of the various antimicrobial susceptibility testing (AST) methods that are currently used or potentially applicable in the foreseeable future, as well as their advantages and disadvantages.

## 2. The Rationale for Performing Susceptibility Testing

The choice of the best therapeutic option for the treatment of bacterial infections relies on the results of AST, a part of the routine work of all clinical microbiological laboratories. These reports provide insight into local patterns of antimicrobial susceptibility, helping physicians to choose the most effective antibiotic therapy [13]. For instance, if the AMR rate of a pathogen is above 20%, that drug should not be administered as a single empiric therapy for infection treatment [14]. Evaluation of the effectiveness of prevention and infection control measures relies as well on the results of AST, e.g., monitoring of resistant pathogens such as MRSA (methicillin-resistant *Staphylococcus aureus*), VRE (vancomycin-resistant enterococci), extended-spectrum beta-lactamase (ESBL)- and carbapenemase-producing *Enterobacterales*, carbapenem-resistant *Acinetobacter baumannii* (CRAB), carbapenem-resistant *Pseudomonas aeruginosa* (CRPA), colistin-resistant bacteria, etc. [15]. Finally, surveillance of antimicrobial resistance is based on routine clinical antimicrobial susceptibility data from microbiological laboratories. Numerous AMR surveillance systems exist, of which the WHOs Global Antimicrobial Resistance and Use Surveillance System (GLASS), European Antimicrobial Resistance Surveillance Network (EARS-Net), and Antibiotic Resistance Laboratory Network (AR Lab Network) of the Centers for Disease Control and Prevention are the most recognizable networks of national surveillance systems providing information on the actual burden of resistance at the international level. Policymakers and health administrators revise the recommendations for empirical treatment for community or hospital-acquired infections according to the local, national, and international AMR data. In addition, prevention and infection control measures are implemented based on the same data as a part of AMS programs [16,17]. Likewise, continuous monitoring provides early warnings of emerging threats and identifies long-term resistance trends.

Although resistance surveillance at the national and international levels is of great benefit to public health, knowledge of the local resistance rates is of even greater practical importance to physicians. An antibiogram represents a convenient and widely available measurement of an institution’s pathogens and susceptibilities [18]. Therefore, it is increasingly suggested that there is the necessity to create local (hospital or institutional) antibiograms specific for each hospital and even ward, annually. This principle applies especially to certain hospital departments where resistance rates are high, such as intensive care units. Additionally, this is particularly relevant for secondary and tertiary hospitals that treat chronically ill patients who have already received multiple antibiotic courses and thus increase antimicrobial selective pressure. Klinker et al. provide the rationale for why hospital AMS programs should implement alternative antibiograms, including combination and syndromic antibiograms, in addition to traditional antibiograms [18]. A combination antibiogram is used to determine in vitro rates of susceptibility to potential antibacterial combination regimens consisting of a first-choice antibiotic plus alternatives. A syndromic antibiogram displays the likelihood of adequate coverage for a specific infection syndrome, considering the weighted incidence of pathogens causing that syndrome. It was developed by Hebert et al. [19] as a weighted-incidence syndromic combination antibiogram. While combination antibiograms are useful in determining combined empiric antibiotic regimens for multidrug-resistant pathogens [20], syndromic antibiograms provide effective antibiotic therapy for a specific infectious syndrome, such as hospital- and ventilator-associated pneumonia [21]. The Clinical and Laboratory Standards Institute (CLSI) has developed guidelines (M39-A4) [22] to provide a standardised template for the preparation of institutional antibiograms. In a retrospective study by Puzniak et al. [23], the utility of combination antibiograms in identifying optimal anti-*P. aeruginosa* drug regimens in US hospitals was evaluated. They found that adding an aminoglycoside to backbone antibiotic, such as extended-spectrum cephalosporin, carbapenem, or piperacillin-tazobactam, resulted in higher susceptibility rates than adding a fluoroquinolone. They concluded that local institutional use of combination antibiograms ensures optimisation and timely administration of appropriate empiric therapy of infections caused by difficult-to-treat pathogens.

Clinical laboratories currently employ several AST methods depending on the equipment and laboratory test menu that they provide. Conventional AST based on phenotypic testing examines the bacterial response in the presence of an antimicrobial agent. Classical culture-dependent methods (e.g., a disk diffusion test, gradient diffusion method) are firmly established in the diagnostic routine, and their main limitation is that the results are obtained for most clinically important bacteria within at least 18–24 h or 48 h, including prior bacterial isolation and identification. The turnaround time is prolonged for anaerobes or some slow-growing fastidious bacteria such as the HACEK group (*Haemophilus* species, *Aggregatibacter* species, *Cardiobacterium hominis*, *Eikenella corrodens*, and *Kingella* species), *Brucella* spp. etc. [24]. For many years, clinical laboratories have been equipped with automated systems based on microdilution trays to provide faster results (6–24 h after initial isolation). However, the time required to obtain the results is similar in comparison with the broth microdilution (BMD) method [25]. Molecular AST is based on the detection of resistance determinants in bacterial isolates or directly in clinical specimens by molecular methods with a turnaround time of approximately 1–6 h [26]. Besides high costs, major drawbacks of molecular methods are detection of the resistance genes targeted only by the known probes and overestimating resistance because the resistance gene is not necessarily associated with the expression of a resistance phenotype. Because of a significant rise in multi- and pan-drug-resistant infections, there is an urgent need for a more rapid and reliable test to improve infection diagnosis and support evidence-based antimicrobial prescribing [27]. The currently used methods for AST are summarised in Figure 1.

## 3. Commonly Used Techniques

### 3.1. Classical Methods

#### 3.1.1. Dilution Methods: Broth Dilution and Agar Dilution

Although new technologies have been introduced to obtain data on bacterial susceptibility to antimicrobial agents, conventional technologies are still in widespread use. Besides disc diffusion susceptibility tests, the most widely used methods include broth macro- and microdilution and agar dilution, representing the reference methods [28]. By using broth and agar dilution methods, the minimum inhibitory concentrations (MICs) of antimicrobial agents (i.e., the lowest concentration at which the agent inhibits the growth of microorganisms) can be determined [29,30]. The MIC value serves as the basis for assessing the susceptibility category of the pathogen to a given antibiotic, of organisms that give ambiguous results, and especially when no clinical breakpoints for disk diffusion are available. Contrary to a qualitative method, the MIC value allows assessing the degree of susceptibility or resistance to the antibiotic [31]. Besides the determination of MICs, the advantage of broth dilution methods is the possibility of obtaining the minimum bactericidal concentration (MBC), which is the lowest concentration of an antimicrobial substance that kills 99.9% of bacteria [32].

The macrodilution method, also known as the in-tube dilution test, uses serial two-fold dilution of antimicrobial substances in corresponding media. A known concentration of suspended bacteria is added to the tubes prepared, as described in [32]. After 24 h of incubation at 37 °C, bacterial growth is measured by turbidity of media, allowing visual determination of MIC values. Another macrodilution method is the time-kill methodology. This test allows monitoring of the effect of different concentrations of antimicrobial substances by examining the rate at which antimicrobials lead to bacterial death—i.e., the bactericidal activity of antimicrobial agents is determined depending on the concentration and time. Bacterial viability is determined by counting colonies on agar plates at regular time points for 24 h [33]. The rate of bacterial growth is monitored via changes in log CFU/mL during the first 24 h time-kill test. Based on the results, experimental curves which represent the absence of growth or the killing effect can be constructed and give us insight into the interaction between the bacteria and the antimicrobial agent. The data can be further analysed using different mathematical models [30,34,35].

The BMD method is standardised, accurate, and inexpensive. Since it is performed in 96-well microtiter plates, it allows the testing of several antimicrobial substances in a row and eight series of two-fold dilutions of antimicrobial agents in one plate. After the dilutions are made, each well is inoculated with standardised bacterial inoculum and incubated for at least 16–24 h. Although this procedure is used as a reference method, it has been improved by the addition of a resazurin colour redox indicator. Resazurin is a blue colour that turns into pink, fluorescent resorufin in the presence of metabolically active bacterial cells. The reduction of resazurin to fluorescent resorufin can be measured fluorimetrically [27,32,36,37,38]. Nowadays, there are several commercially available easy to perform BMD systems such as MBD Sensititre System (Thermo Fisher Scientific, Waltham, MA, USA) and ComASP Colistin (Liofilhem, Roseto degli Abruzzi, Italy), formerly SensiTest Colistin. The MBD Sensititre System can be performed manually or automatically. ComASP Colistin is a compact panel containing the antibiotic colistin in seven two-fold serial dilutions and allows for four samples to be tested simultaneously with the BMD method [39].

The agar dilution method involves adding different concentrations of antimicrobial substances to the non-selective medium before solidification [40]. Afterwards, the standardised bacterial inoculum is spotted on the agar surface. Following overnight incubation, plates are evaluated visually, determining whether growth has occurred at the inoculated sites. The lowest concentration of antibiotics that prevent bacterial growth is considered to be the MIC. This method allows simultaneous testing of different bacterial strains [41].

#### 3.1.2. Antimicrobial Gradient Method

The gradient strip test is a combination of disk-diffusion and dilution method of AST, having advantageous properties of both methods. It allows the MIC to be determined while keeping it simple and easy to use. The method is based on the diffusion of an antibiotic through agar with a continuous gradient. A concordance of the susceptibility categories and MIC values obtained by gradient test and BMD method, a “gold standard” recommended by the European Committee on Antimicrobial Susceptibility Testing (EUCAST) and CLSI, were observed [28,42]. For example, the new ceftazidime–avibactam and ceftolozane–tazobactam gradient tests (Etests, bioMérieux, Marcy-l’Étoile, France) have shown a high categorical agreement between gradient test and BMD, of 96% and 94%, respectively [43,44]. On the other hand, for some antibiotics, such as colistin and tigecycline [45], controversial results have been obtained. Some agar-related factors, i.e., the content of divalent cations, can affect the diffusion of colistin, resulting in false susceptibility. Consequently, BMD remains the only appropriate method for MIC determination for certain antibiotics [46]. Currently, a few commercial gradient strip tests, such as Etest (bioMérieux, France), MIC Test Strip (Liofilchem, Roseto degli Abruzzi, Italy), M.I.C.Evaluator (Thermo Scientific, Waltham, MA, USA), and Ezy MIC Strip (HiMedia Laboratories, Mumbai, India), are available [47]. They can be used for susceptibility testing of microorganisms to antibiotics and antifungals [48,49,50].

A gradient strip test is performed according to the manufacturer’s instructions: a short plastic or paper strip impregnated with antibiotic is placed on inoculated agar (Figure 2). On the standardised 100 mm Petri dish, two strips may be placed, while on the larger 150 mm Petri dish, up to six antibiotics may be tested simultaneously. The MIC of a tested agent is determined by the intersection of a zone of inhibition with the strip and measured using labelled concentrations on the strip. If the intersection is between two values on a scale, a higher value is reported as MIC. In addition, if beta-haemolysis is present on the plate, careful examination of the strip is required since the reporting of the intersection of haemolysis leads to false higher MICs values. Automated systems for reading the results of gradient tests are also available (ADAGIO Automated System, Bio-Rad Laboratories, Hercules, CA, USA).

A variation of gradient tests exists for the detection of various AMR phenotypes. Currently, Etests for phenotypic detection of ESBL production in enterobacteria are available, such as strips with cefotaxime+clavulanic acid, ceftazidime with clavulanic acid, and cefepime with clavulanic acid [51]. The gradient tests for ESBL detection are two-sided strips that contain antibiotic on one end, while on the other is the same antibiotic with clavulanic acid. Reduction in MIC equal to or greater than eight times by the combination of antibiotic and clavulanate refers to ESBL production [52]. Similar to the double-disk synergy test, the phantom zone below the clavulanic end also indicates a positive result. Identification of metallo-beta-lactamase (MBL)-producing bacteria can be carried out using a gradient test. These tests contain carbapenem antibiotic on one side of the strip and the same carbapenem with EDTA on the other side. Imipenem with EDTA for detection of MBL in *Acinetobacter* spp. and *Pseudomonas* spp. is available, although sensitivity and specificity may vary [53,54]. For detection of AmpC beta-lactamase-producing enterobacteria, Etest impregnated with cefotetan on one end and cefotetan–cloxacillin on the other end can be used [55]. Gradient tests with a predefined gradient of vancomycin and teicoplanin on each side of the strip can be used for the detection of glycopeptide resistance in *Staphylococcus aureus* [56]. Since these tests are easy to perform, they could be used as “screening” tests for the detection of emerging resistance patterns among clinically relevant bacteria.

Plenty of advantages of gradient tests are known: simple performance, flexibility in the testing of any combination of antibiotics, and the fact that they do not require expertise and special technologies. Moreover, their use is suited when only a couple of antibiotics are needed to be tested. The price of each strip is relatively high, compared with the price of disks; therefore, gradient tests are usually used to test only a few antibiotics per strain. The incubation length of 16–24 h for gradient tests may represent a disadvantage, as more rapid automated systems are available with the reliable determination of MIC.

#### 3.1.3. Disk Diffusion Test

Since its development in 1940, the disk diffusion (DD) test has remained the most widely used routine AST in clinical microbiological laboratories [57]. It has been standardised to test the susceptibility of the most common and clinically relevant bacteria that cause human diseases [42,58]. The standardisation is a continuous process, and DD for many microorganisms/antimicrobials is an ongoing process [59]. The method is based on placing different antibiotic-impregnated disks on previously inoculated agar with bacterial suspension. The antibiotic diffuses radially outward through the agar medium, producing an antibiotic concentration gradient. After the inhibition zones are established within 24 h of incubation at 35 ± 1 °C, the zone diameters of each tested antibiotic are measured by the naked eye or using an automated system [60]. Obtained results should be interpreted and categorised according to the recommended clinical breakpoint of the standard in use [42,61]. Disk diffusion is the most widely used AST method in microbiology laboratories because of its low cost and ease of performance and applicability of numerous bacterial species and antibiotics [32]. The choice of antibiotic disks is flexible and enables the clinical laboratory to make different combinations according to the bacterial species and the type of sample the isolate was obtained from [62]. Simple interpretation allows the detection of atypical phenotypes and visibility of contamination. However, the main disadvantages are the inability to determine the MIC and delays in getting the results. Reduction in turnaround time and timely treatment are of great importance for critically ill patients. In addition, the biological properties of lag and log phase of bacterial growth and their expression on antibiotic influence should be considered [63]. Nevertheless, methods to reduce incubation time for DD were suggested decades ago [64,65,66]. A revival of that idea led to the development of automated systems (WASPLab, Copan, Murrieta, CA, USA and BD Kiestra, Becton, Dickinson and Company, Franklin Lakes, NJ, USA) for an acceleration of AST by DD [67,68]. The automatisation of the AST by DD leads to a shortening of the required time to obtain results and produce the final report [68]. EUCAST has defined a methodology of disk diffusion rapid AST (RAST), which is performed directly from positive blood culture bottles, with breakpoints for short incubations of 4, 6, and 8 h [69,70,71]. RAST can be implemented in routine laboratories without major investments. The method has been validated for a limited number of bacterial species and antibiotics so far. Furthermore, the combined use of a MALDI-TOF MS for the identification of bacteria and RAST directly from positive blood bottles enables reporting AST results within less than 24 h, which significantly reduced the turnaround time compared with the 24–48 h needed for culturing and classical AST methods, such as DD [72]. The predictive value of direct DD testing from positive blood cultures has been reported to have an important influence on AMS [73].

Quality control testing of media and antibiotics is of great importance for ensuring that disk diffusion is providing accurate and reliable results [59,74]. In some cases, the DD method is more reliable than MIC determination. For instance, in the case of detecting penicillinase-producing *S. aureus* strains, the inhibition zone diameter combined with the EUCAST-based zone edge test is the most sensitive and specific phenotypic method [42,75]. The DD method can be used for screening of susceptibility to a larger number of antibiotics or a whole class of antibiotics; detection of certain important resistance phenotypes, such as ESBL, carbapenemases, inducible resistance to macrolides (Figure 2); or the presence of a heteroresistant population of bacterial species in a sample that cannot be detected by other phenotypic AST methods. The above-mentioned suggests that the DD method will remain a widely used AST method in the future.

#### 3.1.4. Chromogenic Agar Media for Detection of Antimicrobial-Resistant Bacteria

Since the introduction of the first chromogenic media for the detection of antibiotic-resistant bacteria 20 years ago, a variety of different media for the detection of clinically important resistant pathogens—such as MRSA, VRE, and ESBL- and carbapenemases-producing or colistin-resistant Gram-negative bacteria—have been developed [76,77,78,79,80].

The main purpose of the development of chromogenic media was to enable more rapid detection and identification of resistant microorganisms. The target organisms are characterised by specific enzyme systems that metabolise the substrates to release the chromogen. The chromogen can then be visually detected by direct observation of a distinct colour change in the medium. Thus, these selective and differential media enable target pathogens to grow as coloured colonies. Compared with the use of conventional culture media, the use of chromogenic agar often reduces the costs and labour time [81]. Their primary use is for screening of patients colonised with various pathogens, and therefore they are valuable in infection prevention [82] and control of hospital-acquired infections. The sensitivity and specificity of chromogenic media depend on the manufacturer and the type of microorganism detected; thus, additional identification confirmation of the resistant bacteria is sometimes needed. Because of their wide applicability, new chromogenic media are being developed [83,84].

#### 3.1.5. Colourimetric Tests for Detection of Antimicrobial-Resistant Bacteria

Colourimetric tests represent phenotypic methods developed for the detection of AMR. They are based upon the bacterial enzymes hydrolysing test component, which is detected by the changes in pH values and the colour of chromogenic substances. Briefly, bacterial suspension or bacterial lysate suspension is added to a detection solution containing antibiotic and pH indicator dyes, such as phenol red, and incubated for a short period of time, no longer than a couple of hours. The pH of the detection solution changes due to the growth of antibiotic-resistant bacteria, or bacterial enzymes activity, and subsequently, the colour of the solution changes, which can then be visually observed. These tests were shown to be fast, easy to perform and interpret, and highly sensitive and specific. A good example of such colourimetric tests is Carba NP (bioMérieux, Marcy-l’Étoile, France), which detects carbapenemase-producing bacteria. The test gives reliable results in 30 min to 2 h, making it the quick and easy way to control carbapenemase producers [85].

## 4. Current Technologies for Rapid AST

### 4.1. Automated and Semi-Automated Devices Based on Microdilution Susceptibility Testing

Clinical microbiology laboratories are under increasing pressure to provide fast and reliable microbial identification (ID) and AST [86]. Automated and semi-automated devices for bacterial ID and AST are worthy of the task and have significantly improved laboratory efficiency. Nowadays, automation has been successfully implemented in most clinical microbiological laboratories to reduce turnaround times, increase efficiency, and improve cost-effectiveness [86,87]. Various test systems—such as the VITEK 2 (bioMérieux, France), MicroScan Walkaway (Dade-Behring MicroScan, Deerfield, IL, USA), and Phoenix system (BD Diagnostic Systems, Baltimore, MD, USA)—have been widely used over the last decades. These instruments, using optical systems for measuring subtle changes, determine bacterial growth and antimicrobial susceptibility [88] and can produce results in a shorter time (6–12 h) than conventional manual assessment [36].

VITEK 2 Systems—The first generation of VITEK system with a turnaround time of 13 h was developed for enumeration and identification of bacteria and yeasts in 1973. The VITEK 2 System, the next-generation of an instrument, is a BMD-based AST system that uses 64-well plastic cards containing 17–20 antimicrobial agents. If the bacterial isolate is not previously identified, one card is used for bacterial identification (ID card) and the other for antimicrobial susceptibility testing (AST card). Two Vitek 2 instruments are available with test card (ID and AST) capacities of 60 cards (Vitek 2) and 120 cards (Vitek 2 XL). Results are reported in 4–18 h, containing MIC and category of susceptibility, whereas the detection of AMR is facilitated by the Advanced Expert System (AES). The currently available Vitek 2 Compact instruments can use 15, 30, and 60 cards. The main advantage of the Vitek 2 system with computer software is the determination of susceptibility of clinically important resistant pathogens, such as *Staphylococcus aureus* and *Enterococcus faecalis*, to an additional four to ten antibiotics [86,89,90].Phoenix System—The Phoenix System is widely accepted and used in clinical microbiology laboratories for identification testing (ID) and antimicrobial susceptibility testing (AST). The principle of determining the susceptibility is based on the use of an oxidation-reduction indicator (resazurin dye or Alamar blue) and the detection of bacterial growth in the presence of various concentrations of the antimicrobial agent. In the Phoenix instrument, a maximum of 100 tests can be performed by using Phoenix ID/AST combination panels (51 for ID and 85 for AST). The instrument performs automatic reading at 20 min intervals during incubation for up to 18 h and provides accurate and rapid susceptibility results with easy workflow for the laboratory worker. In 2014, the new panel for susceptibility of Gram-negative bacteria was introduced for the Phoenix system to be used in combination with the BD Bruker MALDI-TOF [91].MicroScan WalkAway plus System—The MicroScan WalkAway plus System provides accurate and rapid identification and susceptibility results for a wide range of Gram-positive and Gram-negative aerobic bacteria. The instrument utilises three types of panel configurations: combo panels, breakpoint combo panels, and MIC panels. There are two types of system: 40- and 96-panel capacity models. The panels are manually inoculated, rehydrated by the RENOK inoculator, and read automatically. The results are obtained after 4.5–18 h by reading of rapid panels [91].MicroScan AutoScan 4—The AutoScan 4 is a semiautomated instrument mostly used in smaller laboratories or for the testing of supplemental antimicrobial agents. The instrument provides simplified ID/AST testing in a highly reliable and affordable package. The system uses the off-line incubation of the conventional MicroScan AST panels. The panels are manually inoculated or with the MicroScan Renok instrument and read automatically [91].MicroScan WalkAway System—The first generation of the MicroScan WalkAway System available on the market is the AutoSCAN-3. The new versions of instruments Auto-ACAN-4 and AutoSCAN-WalkAway are improved and use dry panels that do not need refrigeration. The AutoSCAN-WalkAway system detects bacterial enzymatic activity and can process 96 panels at the same [86].

Each of the above-mentioned systems has inherent advantages and limitations, and the results vary widely by antimicrobial drugs, software versions, and cards used. Hence, some of the systems are not reliable for correct categorisation of susceptibility profiles for certain drugs, leading to wrong classifications of susceptibility categories [92]. It seems that low inoculum size has a major influence on the outcome of these systems, with false susceptibilities being reported. Additionally, software updates and synchronisation of breakpoints according to the current standards are mandatory. Thus, it is incumbent upon the instrument manufacturer to keep pace with the breakpoint updates and make relevant improvements, such as extending the detection limit and verifying the performance of the AST system with the revised breakpoints internally, to avoid the problem of uncategorised results [93]. Panels usually contain only several concentrations of each antimicrobial agent, and the resulting MIC is not always given as an exact value. In contrast, classical BMD contains a wide range of doubling dilution antimicrobial concentrations for the determination of the MIC, thus obtaining the more precise value. In addition, according to the previously published reports, many of the resistance phenotypes are not easily detected using the automated susceptibility testing methods so prevalent in today’s clinical laboratories [94]. Nonetheless, interestingly, the ability of automated systems to detect inducible resistance to clindamycin in 524 isolates of *Staphylococcus* spp. revealed sensitivity and specificity of 100% and 99.6%, respectively, for Phoenix, and 91.1% and 99.8%, respectively, for Vitek 2 [95]. The multicentre evaluation showed that categorical agreement between the Phoenix system and a BMD reference method for 2013 streptococcal isolates including *Streptococcus pneumoniae*, viridans group streptococci, and beta-haemolytic *Streptococcus* groups A, B, C, and G ranged from 92% to 100%, with one exception for viridans streptococci and penicillin, which was 87% [96]. However, according to the results of the evaluation of ASTs obtained using Vitek 2, Phoenix, and MicroScan, caution should be taken for AST of *Stenotrophomonas maltophilia*, as a high rate of errors may be observed [97].

### 4.2. Molecular-Based Techniques for Resistance Detection

Molecular AST directly detects specific resistance genes, as well as mutations in and expression of these genes. These molecular methods have been developed and tested as an alternative for or complementary to conventional AST and are generally faster than classic culture-based assays, with the test results available within one to a few hours [98] (Figure 3). Most of the molecular AST methods fall into one of the three categories: amplification-based, hybridization-based, or sequence-based. In amplification-based methods, the target gene sequence is amplified to allow detection; in hybridization-based techniques, hybridized nucleic acid probes target gene sequences allowing detection; and in sequence-based approaches, genome sequences are analysed to detect resistance-conferring mutations or resistance genes.

#### 4.2.1. Polymerase Chain Reaction

The most widely used nucleic acid amplification-based method for the detection of specific resistance genes is polymerase chain reaction (PCR). Both real-time and conventional PCR rely on the amplification of nucleic acid sequences that encode resistance to an antibiotic. New PCR-based methods are being developed for the detection of genetic determinants of resistance to a variety of antibiotics for various bacterial species, as our knowledge about the genetic basis of antibiotic resistance increases [99]. Multiplex assays for simultaneous testing of multiple genetic determinants in various bacterial species have also been developed, i.e., multiplex assays for identifying numerous cephalosporinase- and carbapenemase-encoding genes, such as *bla_KPC_*, *bla_NDM_*, *bla_IMP_*, *bla_VIM_*, *bla_AmpC_*, *bla_TEM_*, *bla_SHV_*, and *bla_OXA_*, or *mecA* gene-encoding methicillin resistance in MRSA [100,101]. OpGen, Inc. (Rockville, MD, USA) has recently released the multiplex-based Acuitas^®^ AMR Gene Panel that detects 28 genetic AMR markers, covering select drugs in nine classes of antibiotics, from 26 different pathogens [102]. The advantage of this test in comparison with other commercially available molecular tests is that it also detects non-beta-lactam resistance genes and those for what would be considered “last-resort antibiotics”, such as colistin.

Real-time PCR (quantitative PCR, qPCR) is one of the most ubiquitous methods found throughout clinical microbiology. Although costlier, qPCR offers several advantages over conventional PCR, including the measurement of data in real-time, greater sensitivity, reduced risk of carryover contamination, and greater amenity to multiplexing. Further advantages are that many systems are partially or even completely automated, such as GeneXpert^®^ Instrument Systems (Cepheid Corp., Sunnyvale, CA, USA) and BD MAX System platform (Becton Dickinson, Franklin Lakes, NJ, USA), which are easily operated and can be used for the detection of carbapenemases, ESBLs, MRSA, VRE, etc. [103,104,105]. The downside is that they are limited to using test assays only from specific manufacturers, with GeneXpert^®^ Instrument Systems requiring GeneXpert assays (Cepheid Corp., Sunnyvale, CA, USA) and BD MAX System using Check-Points^®^ qPCR assays (Wageningen, The Netherlands). Tests based on qPCR can also be used for phenotypic differentiation of resistant and susceptible strains due to its ability to measure genome copy numbers during bacterial growth in the presence of antibiotics [106]. The major disadvantage is that the system cannot provide information about the mechanism of resistance and that it requires the previous culture, meaning that the primary clinical samples cannot be used.

#### 4.2.2. DNA-Microarrays

DNA-microarrays are used to identify the presence of specific nucleic acid sequences using complementary short oligonucleotides immobilised on a solid surface. Since these oligonucleotides can be assembled onto solid surfaces in close proximity, this method could detect numerous sequences in a single assay, which would allow simultaneous, in parallel detection of different pathogens and detection of vast numbers of different resistance genes, as well as detecting numerous distinct mechanisms of resistance or variants of a single mechanism present in bacterial isolates, as opposed to PCR-based approaches [107]. The Verigene system (Luminex Corporation, Austin, TX, USA) has developed Blood Culture Multiplex Microarray-Based Molecular assays for rapid diagnostics of 12 Gram-positive and 9 Gram-negative bacteria, along with their associated resistance genes (i.e., *mecA*, *vanA*, *vanB*, *bla_KPC_*, *bla_NDM_*, *bla_IMP_*, and *bla_VIM_*) [108]. In addition to qPCR-based assays, Check-Points^®^ has also developed the CHECK-MDR CT103 DNA microarray for the detection of the clinically most prevalent ESBLs and carbapenemases, as well as mobile colistin resistance (*mcr*) genes in Gram-negative bacteria [109]. Both of these microarray tests have shown high sensitivity (94–100%) and specificity (94–98%), and CHECK-MDR CT103 DNA microarray also showed the ability to discriminate between carbapenemase and ESBL variants of GES-type beta-lactamase [108,109].

The advantages of currently available molecular-based methods are that they are direct, rapid, highly sensitive, and specific, thus potentially allowing the earlier administration of targeted therapy [98]. Furthermore, for some methods, direct clinical samples can be used. However, it should be noted that the presence of a resistance marker does not always have to correlate with phenotypic resistance. Additionally, the extent and intensity of gene expression are important parameters, as some genes need different expression levels to produce resistance. A potential solution to this issue would be the use of reverse transcription qPCR, which relies on the measurement of gene transcripts (RNA levels) instead of the presence of a gene [110,111]. Another drawback is that these methods can only detect resistances that are searched for, and not novel or uncharacterised mechanisms of resistance, which could lead to false-negative results and inappropriate classification of resistant isolates as susceptible. A final consideration is that these methods are not capable of defining MIC values. As such, these methods have to be validated against phenotypic data to be useful, and extensive resistance marker databases and innovative bioinformatics methodologies are mandatory requirements. Nevertheless, molecular-based AST methods are a safe, efficient, and reliable screening tool in clinical settings. As experience with these tests grows, and as data are gathered on their efficacy and clinical impact, they will likely be more widely adopted.

#### 4.2.3. Whole-Genome Sequencing in Antimicrobial Susceptibility Testing

As DNA sequencing technology and bioinformatics pipelines for genome assembly and analysis advance, the possibility of using these techniques for the detection of antibiotic resistance opens. Applying whole-genome sequencing (WGS) would essentially enable the detection of all genes involved in AMR, which would help make comprehensive databases of all species-specific resistance factors (i.e., CARD-Comprehensive Antibiotic Resistance Database—https://card.mcmaster.ca, ResFinder—https://cge.cbs.dtu.dk/services/ResFinder (accessed on 27 February 2022)) and make in silico AMR detection possible. Recent studies showed high concordance between the resistance profiles obtained using WGS and those obtained using phenotypic susceptibility testing, demonstrating that data obtained from genome sequences can correlate well with phenotypic resistance in some cases [112,113]. In addition to genome-based resistome analyses, RNA-mediated transcriptomic approaches have also been described [114,115]. Despite all of the advantages, WGS is not routinely performed in clinical practice. Considering the turnaround times of WGS, the existence of unknown resistance mechanisms, and the elevated cost compared with traditional and emerging techniques, the use of WGS for AST is not yet part of routine practice in clinical microbiology [116,117].

### 4.3. Mass Spectrometry

Matrix-assisted laser desorption ionization-time of flight mass spectrometry (MALDI-TOF MS) was discovered in the 1980s and introduced into the microbiological routine as an effective tool for bacterial and yeast identification about 15 years ago. It has been applied to classify the specific bacterial protein contents and their matching protein biomarkers because of its rapid turnaround time, low sample volume requirements, and per-sample costs [118]. Several MALDI-TOF MS-based methods have been proposed for rapid detection of antimicrobial resistance, including monitoring antibiotic modification by bacterial culture (e.g., beta-lactam hydrolysis [65,119]), acetylation of fluoroquinolones [120], direct detection of proteins involved in specific resistance mechanisms [121,122], and detection of stable isotope labelling that requires expensive, isotopically labelled media [123,124]. The hydrolysis of the target beta-lactam antibiotic, as shown by peak disappearance, is used to detect beta-lactamase-producing bacteria using MALDI-TOF MS. As a result, the assay for detecting carbapenemase production [125] automatically determines sensitivity or resistance depending on the degree of antibiotic hydrolysis. The method had 98% sensitivity and 100% specificity after 30 min of incubation of bacteria with the antibiotic, with both reaching 100% after 60 min of incubation [126,127]. However, beta-lactam resistance is only recognized when it is mediated by beta-lactamases; alternative resistance mechanisms have not been elucidated; therefore, other tests should confirm negative results. Another assay for the detection of carbapenemases is a rapid and novel method using detonation nanodiamonds (DNDs) as a platform for the concentration and extraction of *A. baumannii* carbapenemase-associated proteins before MALDI-TOF MS analysis [128]. The sensitivity and the specificity of the proposed platform could reach 96% and 73%, as compared with traditional imipenem susceptibility testing, and 100% compared with PCR results. This method may detect the carbapenemases produced by *A. baumannii* in 90 min and does not require the addition of a carbapenemase substrate, as other mass spectrometric methods do. It is efficient for detecting other carbapenemase-producing bacteria. 

MALDI Biotyper-Antibiotic Susceptibility Test Rapid Assay (MBT-ASTRA) is an alternative MS-based method for AST which utilises semi-quantitative MALDI-TOF MS to measure the relative growth rates of bacterial isolates exposed to antibiotics compared with untreated controls during a short incubation step. A software tool calculates and compares the area under the curves (AUCs) of spectra of bacteria either exposed or not to an antibiotic [129]. In this method, if the microbial strain is susceptible, the AUC of the bacterial suspension with the antibiotic will be reduced compared with that without antibiotics, whereas with a resistant strain the AUCs with or without antibiotics will be comparable. The main advantage of the MBT-ASTRA is that the assay does not depend on the resistance mechanism and is utilisable with any antibiotic [130]. Moreover, it does not require specialised media or instrumentation, beyond the MALDI-TOF mass spectrometer. However, a drawback of the MBT-ASTRA assay is that the concentration of antibiotics used and the incubation time must be optimised for each species and antibiotic combination [129].

MBT-Resist assay, based on the detection of peak shift after stable isotope labelling, is an approach that uses the following principle: bacteria are grown in parallel in two distinct culture mediums, one containing 12C as a carbon component and the other containing 13C [131]. The system compares the mass spectrum of bacteria grown on an isotope-labelled medium with antibiotics to the mass spectrum of the same strain grown on an unlabelled medium without antibiotics. Resistant strains can thrive in the presence of antibiotics, incorporating 13C into the polypeptide, causing a shift in the peak to a higher *m*/*z* in the mass spectrum [131]. 

Antibiotic resistance by direct-on-target microdroplet growth assay (DOT-MGA) is a novel approach for detecting antimicrobial susceptibility in bacteria treated with breakpoint concentrations of antibiotic on the target plate of MALDI-TOF MS [132]. The best performance was obtained by recovering bacteria from positive blood cultures and after a 4 h incubation of microdroplets with or without meropenem at the breakpoint concentration. Under these conditions, 96.3% validity, 91.7% sensitivity, and 100% specificity were achieved. Recently, a screening panel for the detection of ESBL and AmpC beta-lactamase activity was developed [133]. Compared with the PCR results, positive percentage agreement values for ESBL, AmpC, and ESBL + AmpC resistance were 94.4%, 94.4%, and 100%, and negative percentage agreement values were 100%, 93.7%, and 100%, respectively. The accuracy of the DOT-MGA achieved results incomparable with those of the BMD assay, with a time saving of about 14 h, and higher than combination disk tests.

According to Yoon et al., due to the great speed and simple application, MALDI-TOF MS would be the most suitable for endemic AMR clinical strains in specific settings, i.e., MRSA, VRE, CRAB, CRPA, and ESBL-, AmpC-, and carbapenemase-producing Enterobacterales [134].

The advantages and disadvantages of the commonly used methods of antimicrobial susceptibility testing were summarised in Table 1.

## 5. Selection of Antimicrobial Drugs for Susceptibility Testing, Interpretation, and Reporting

Since there are a large number of antibiotics in use, it would be irrational and pointless to test the susceptibility of isolates to all of them. Although each laboratory should establish its own panels of antibiotics depending on the bacterial isolate, for which susceptibility is tested, there are some common rules regarding the choice of antibiotics [36]. The number and selection of antimicrobials tested are primarily dependent on the organism isolated, infection site, type of infection (community or hospital-acquired), comorbidities, patient’s age, and gender, but also the institution’s formulary, physician requests, and the automated panel or other testing methodology used [135].

Interpretation of AST results and reporting of bacterial susceptibility categories to antibiotics is based on the breakpoints published by the two most commonly used systems worldwide: CLSI and EUCAST [42,61]. CLSI provides recommendations for agents that are important to test routinely (group A) and those that may be tested or reported selectively based on the institution’s formulary (groups B and C) [136]. Accurate identification of bacteria is crucial in the choice of antibiotics [137]. It is well known that resistance mechanisms have not been observed in some bacterial species so far, e.g., continued penicillin susceptibility of *Streptococcus pyogenes* [26]. 

It is also known that some bacterial species are intrinsically resistant to particular antibiotics or antibiotic classes. It is systematically presented in the EUCAST document “Intrinsic Resistance and Unusual Phenotypes” [138]. Therefore, it is unnecessary to test certain bacterial species for activity in vitro, but microbiologists should report intrinsic resistance to clinicians. This document also indicates when to avoid the use of antimicrobial agents that are likely to result in treatment failure. 

Exceptional or unusual resistance phenotypes are also described in the EUCAST document [138], and knowledge of those phenotypes allows microbiologists to further investigate the obtained results. Examples of such phenotypes are penicillin-resistant group A streptococci or vancomycin-resistant staphylococci. These results should be confirmed and checked before reporting. Organisms with unusual phenotypes could be seen due to the emergence of antibiotic resistance, but it is more likely that an error occurred during the strain identification, antibacterial susceptibility testing, or when the mixed culture was tested [139].

Due to the similar in vitro activities of some antimicrobials belonging to the same class, a representative antibiotic should be tested to predict susceptibility to other class members [27]. Thus, for example, in beta-haemolytic streptococci, the susceptibility to penicillins can be inferred from benzylpenicillin susceptibility results [42]. Susceptibility of staphylococci to beta-lactams can be assessed by susceptibility to cefoxitin, while susceptibility of staphylococci and streptococci to macrolides and fluoroquinolones could be predicted using erythromycin and norfloxacin, respectively [135]. In enterococci, ampicillin is used as an indicator of resistance to ampicillin, amoxicillin, and piperacillin with and without beta-lactamase inhibitor. It is not commonly used in Gram-negative bacteria due to their more heterogeneity in resistance.

Until recently, both EUCAST and CLSI use “susceptible” (S), “intermediate” (I), and “resistant” (R) categories and also shared their definitions. While categories S and R still do not differ in interpretation in both standards and are easily understood, category I was not clear since it represented few definitions into one: uncertain therapeutic effect, appropriate to use when antibiotic is physiologically concentrated at the site of infection, appropriate to use a high dosage of the drug, etc. [140]. The above-described interpretation of the category I led clinicians to either avoid or misuse certain antibiotics [140]. Consequently, in June 2018 the EUCAST Steering Committee changed the definitions of AST categories to susceptible, standard dosing regimen (S), susceptible, with increased exposure (I), and resistant (R) [141]. This is done to emphasise the close relationship between the susceptibility of the organism and the exposure of the organism to antibiotics at the site of infection [142], encouraging physicians to use antibiotics from the I category at the appropriate dosage. EUCAST breakpoints are based on dose and mode of administration as indicated in rationale documents and files of the recommended AST breakpoints [143]. Tables of recommended doses for antibiotics, standard dose regimen for S, and high doses for the new I category were also published [42]. This relatively new rule still needs to be implemented in healthcare professional training and clinical use [31].

Antimicrobial susceptibility categories help clinicians to predict the outcome of antibiotic treatment and provide them information on the recommended dosage. For some infections that require high doses of antibiotics for reaching therapeutic concentrations (e.g., meningitis), EUCAST listed meningitis breakpoints that are different from breakpoints for non-meningeal isolates [144]. Furthermore, for the treatment of infections caused by *Pseudomonas* spp., an increased antibiotic concentration on the site of infection is required. Therefore, according to the EUCAST recommendations, the majority of isolates belong to the category “Susceptible, increased exposure” when evaluating susceptibility to fluoroquinolones, penicillins, and third and fourth generations of cephalosporins, aztreonam, and imipenem [42].

The principle of the automated systems for AST is to interpret the results of MICs to the susceptibility category that corresponds to built-in standards (e.g., EUCAST, CLSI). This allows concurrent reporting of categories of susceptibility, values of MICs, and even some resistance phenotypes, providing valuable information for patient treatment. It also allows clinicians and pharmacologists to determine the appropriate dosage for each patient, which represents an individual or personalised medicine. Such an approach enables the estimation of the degree of resistance or susceptibility to the tested agent [31]. However, this assumption should not be based on a direct comparison of the MIC values obtained during testing, nor under the assumption that lower MIC values indicate greater sensitivity of tested bacteria in all cases. Additionally, MICs values can indicate lower effectiveness of the antibiotic, despite the susceptibility category. For example, the MIC value of 2 mg/L for vancomycin predicts the risk of treatment failure in tested *S. aureus* strains, even though it is still in the susceptible category [145]. 

Although both EUCAST/CLSI standards are constantly being improved and published annually, there are still limitations in reporting. Thus, interpretations of susceptibility to some common pathogens (e.g., *Acinetobacter* spp. and *Stenotrophomonas maltophilia*) and key antibiotics are lacking, resulting in the absence of AST reporting. 

Overall, the general recommendation is to report susceptibility to narrow-spectrum antibiotics whenever possible and to avoid reporting of broad-spectrum antibiotics that pose higher AMR selective pressure (carbapenems, fluoroquinolones) [146,147]. Selective reporting of AST results is recognized as a useful tool for antibiotic stewardship. It implies performing AST according to usual practices, but the results are reported to the physician only for a few antibiotics recognised as a first-line choice [14]. Therefore, microbiologists play an essential role in AMS [148,149]. As Tebano G et al. [14] propose, selective reporting might achieve the following: firstly, reducing unnecessary antibiotic prescriptions in cases of the detection of contaminants or members of the normal microbiota. Secondly, reducing inappropriate antibiotic prescriptions by persuading clinicians to choose narrow- over broad-spectrum antibiotics. Both goals could be obtained by reducing the number of reported antibiotics or selective exclusion of certain antibiotics from the reporting where appropriate. Some countries established their National Antimicrobial Committee to address the challenges of AMR [150,151]. These committees adapt the recommendations of EUCAST/CLSI in accordance with the epidemiological specificities of their countries and provide laboratory guidelines for antibiotic selection, testing, and reporting for their national-level use. In conclusion, evidence-based data shows that selective reporting leads to improved antibiotic usage, reducing both unnecessary and inappropriate prescribing. It can be concluded that, although numerous interventions have been proposed to reduce antibiotic consumption and bacterial resistance, much effort is still needed, especially in middle- and low-income countries. Therefore, wider and more consistent support in the implementation of these strategies is highly needed on a global scale [152].

## 6. Quality Assurance in Antimicrobial Susceptibility Testing

Soon after the introduction of AST in clinical laboratories, the need for standardization of the process was recognized. In the past, several of the European national antimicrobial breakpoint Committees developed their own AST standards. Consequently, the national Committees initiated the standardisation of AST performance and interpretation according to the EUCAST recommendations [153]. A much older and widely used standard is the American CLSI, formerly NCCLS [61]. Although the process of harmonization of these two organizations is ongoing, there are still significant differences.

All AST needs to be evaluated in order to provide a certain quality of the results. For conventional methods, manufacturers of antibiotic disks or tablets, gradient tests, and commercial microdilutions plates with a predetermined concentration of antibiotics have to provide a certain quality of the products, as do suppliers of automated or semi-automated tests. In addition, the microbiology laboratories are responsible for the adequate performance of the antimicrobial susceptibility tests. The control of the quality of performance is ensured through internal and external controls. Internal quality assessment should be provided daily or less frequently if the performance is consistent. External quality assessment tests the overall performance of the laboratory by a collection of bacterial strains of undisclosed susceptibility, usually sent by central international laboratories such as NEQUAS in the UK. After the results are sent back, a detailed evaluation of the participating laboratory performance is provided by the central laboratory [154]. 

Quality control is responsible only for the analytical phase of susceptibility testing, monitoring only the performance of the used test(s). The troubleshooting algorithm for AST QC is shown in Table 2, adapted from the previously published report [155].

Quality assurance (QA), on the other hand, provides the proper performance of all three phases of testing: pre-analytical, analytical, and post-analytical. Major components of the QA program for susceptibility testing are the following: clinically relevant testing strategies, testing of reference quality control (QC) strains, technical competency, organism antibiogram verification, supervisor review of results, procedure manual, proficiency surveys. Quality control includes the following: the precision (repeatability) and accuracy of AST procedures, the performance of reagents used in the tests, the performance of persons who carry out the tests and read the results [156].

The AST procedures include diffusion susceptibility methods, dilution susceptibility testing, and different automated systems. The results of commercial automated systems are obtained more rapidly and are highly reproducible; the use of the automated system software supports the interpretation according to different standards and expert rules. Unusual resistance phenotype or resistance types that are difficult to detect may represent a challenge [36].

As for automated antimicrobial susceptibility systems, which are used more and more frequently in microbiological laboratories, usually, every device has its QC procedure. The most commonly used automated systems for antimicrobial susceptibility are VITEK 2 System (bioMérieux, France), BD Phoenix System (BD Diagnostic Systems, Baltimore, MD, USA), MicroScan WalkAway SI (Siemens Healthcare Diagnostics, West Sacramento, CA, USA), and TREK Sensititre (ARIS 2X, Trek Diagnostic Systems, Cleveland, OH, USA). Quality controls of such devices are not included in standards such as EUCAST or CLSI [61,156,157]. Agencies that provide the clearance of commercial systems for general use, such as the FDA for the USA, should use reference methods before the clearance for use is given. 

To evaluate or compare different susceptibility tests, usually these types of errors are considered: very major error, major error, and minor error. Very major error refers to the characterisation of the resistant isolate as susceptible. A major error is made when a susceptible isolate is characterised as resistant. When an intermediate isolate is characterised as susceptible or resistant, as well as when susceptible or resistant isolate is reported as intermediate, a minor error is made. Errors in antimicrobial susceptibility testing should be monitored by laboratories and carefully analysed. Potential problems either with the identification or susceptibility testing may be revealed [158]. 

For evaluation of new susceptibility tests or devices, a certain number of susceptible as well as resistant strains should be tested against different antibiotics. Additionally, all types of the above-mentioned errors should be evaluated [36]. For new antimicrobial susceptibility devices, acceptable performance according to FDA considers that major errors should not exceed 3% based on the number of susceptible organisms tested [159]. The acceptable numbers of very major discrepancies rate should be 7.5% or less (for the upper 95% confidence limit) and 1.5% or less (for the lower 95% confidence limit), based on the number of resistant organisms tested. Additionally, growth failure rates in the system have to be less than 10%.

The above-mentioned criteria may not be identical for acceptable accuracy compared with the international standards on susceptibility test device evaluation [160].

## 7. Near-Future Perspectives for Antimicrobial Susceptibility Testing

In the last decades, several innovative approaches for AST have been developed. Some of the most promising platforms for the forthcoming period—MALDI-TOF, flow cytometry, and isothermal microcalorimetry—will be briefly described. 

Application of MALDI-TOF MS as routine rapid AST requires additional validation, such as standardised protocols, test kits, and software [161]. Further research efforts are needed to refine and optimise MALDI-TOF MS-based assays to obtain accurate and reliable results in the shortest possible time. A major focus of future research in this field will be to achieve the standardisation of methods and simultaneous susceptibility testing of microbes to various classes of antimicrobials [162]. So far, only two commercially available kits with software for automated interpretation of spectra have been authorised in Europe to detect carbapenemase activity or resistance toward third-generation cephalosporins in clinical microbiology laboratories.

By using flow cytometry, the changes in morphology, physiological and metabolic activity, and survival of microorganisms can be monitored after exposure to antimicrobials. The effects of treatment can be seen within a few hours, suggesting a strong potential for AST. Using nuclear dyes that do not penetrate the cell walls of healthy organisms, the amount of dying and dead cells can be rapidly determined. In order to determine the emission spectrum, the cells must go through a flow channel and be excited by a laser to release dye [163]. Nevertheless, the future challenge is to improve the ability of a system to distinguish cellular damage caused by -cidal vs. -static antibiotics, as well as to address the issue of autofluorescence of certain species of bacteria. In addition, an enormous amount of work is required for the verification of the clinical database and the method itself.

Isothermal microcalorimetry is a dynamic technique allowing the measurement of heat generated by the metabolism of actively growing cells. This technique is not new; however, it has been successfully adapted for AST of bacteria [164]. Additional advantages of this technique are that it requires a small culture volume, the testing is performed in sealed ampules, and monitoring during testing does not require manual manipulation. The completed analysis can also provide information on the -static vs. -cidal activity of an agent.

Up to now, various alternative approaches to combat AMR have been proposed. For instance, epigenetic agents in modulating antibiotic susceptibility [165], agents that inhibit virulence-conferring factors such as multidrug efflux inhibitors [166], or targets’ essential processes for bacterial survival, such as cell division inhibitor [167], have been recently developed. If such antibacterial agents become licensed for treatment, antimicrobial susceptibility techniques should be updated along with the new strategies developed to counter AMR.

## 8. Conclusions

The timely administration of appropriate antimicrobial therapy based on accurate AST is widely recognised as a cornerstone of the management of infectious diseases. Despite the shortcomings of the traditional AST methods, such as BMD and DD, their use is required in clinical practice to obtain the correct results according to the standardised protocols of EUCAST and CLSI, or for comparison against the results of novel techniques. Molecular-based methods are rapid, efficient, and reliable, with high specificity and sensitivity. Nonetheless, additional efforts are needed to bring closer NGS technology to routine microbiology. MALDI-TOF MS appears to be a promising highly specific system with reduced time and low cost of consumables. However, this method is not yet validated for all species and antimicrobials, and the acquisition costs of the system and its maintenance are high. Hence, further improvement in the currently used and novel AST methods and instruments is mandatory for speeding up the determination of antimicrobial efficacy in clinical microbiology laboratories in the foreseeable future.

## Figures and Tables

**Figure 1 antibiotics-11-00427-f001:**
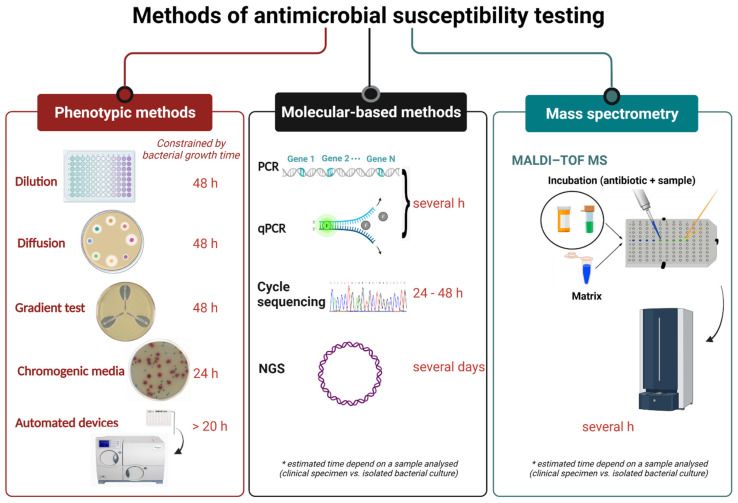
Current methods for antimicrobial susceptibility testing and turnaround time (created with BioRender.com, accessed on 27 February 2022. Reproduction of this figure requires permission from BioRender.com). PCR—polymerase chain reaction. qPCR—quantitative polymerase chain reaction. NGS—next-generation sequencing. MALDI-TOF MS—matrix-assisted laser desorption/ionization time-of-flight mass spectrometry.

**Figure 2 antibiotics-11-00427-f002:**
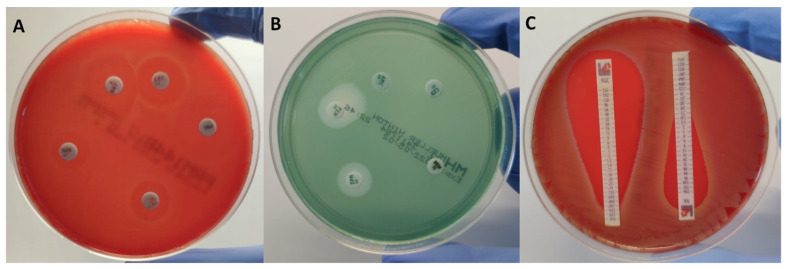
Disk diffusion and gradient test of various bacterial isolates. (**A**)—Antimicrobial susceptibility of *Streptococcus pyogenes* showing iMLS phenotype, using disk diffusion method. (**B**)—Antimicrobial susceptibility of extended-spectrum beta-lactamase-producing *Pseudomonas aeruginosa*, using disk diffusion method. (**C**)—Gradient test of *Enterococcus* spp. iMLS phenotype—inducible macrolide, lincosamide, and streptogramin phenotype.

**Figure 3 antibiotics-11-00427-f003:**
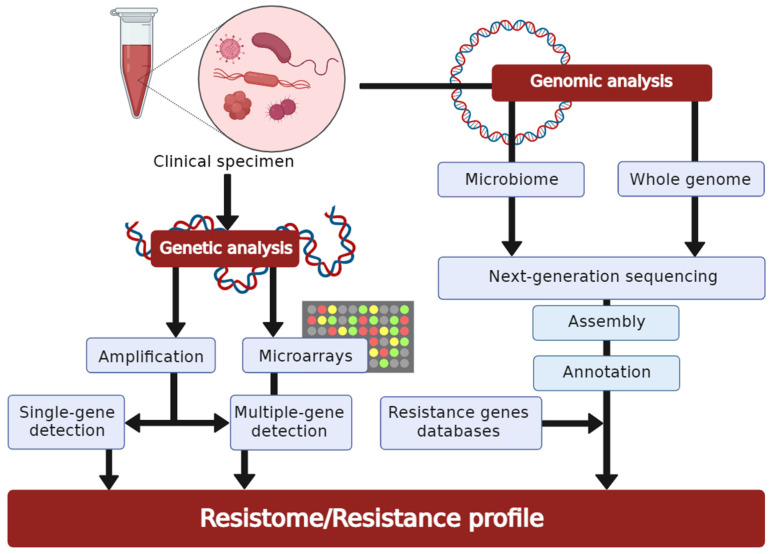
The basic workflow of molecular-based techniques for antimicrobial susceptibility testing. The routes from a clinical specimen to a final result are indicated by arrows (created with BioRender.com, accessed on 27 February 2022. Reproduction of this figure requires permission from BioRender.com).

**Table 1 antibiotics-11-00427-t001:** Advantages and disadvantages of the common methods of antimicrobial susceptibility testing.

Method	Advantage	Disadvantage	Comments
Broth dilution	Well-standardised	Time-consuming	Quantitative **
	Harmonised	Individual mistakes	
	Commercially available tests are easy to perform		
Agar Dilution	Well-standardised	Time-consuming	Quantitative
	Suitable for testing a large number of isolates	Limited concentration of antimicrobial agents	Possible automation in part
Disk diffusion	Simple to perform	Time-consuming	Qualitative *
	Low cost	No MIC value	
	Simple and fast interpretation	The inability for some antibiotics to be tested	
	The high number of test antibiotics per test		
	High flexibility in antibiotic selection		
	Detection of resistance patterns		
	Mass use and the possibility of automatisation		
	A number of a different use (AST, identification, screening, etc.)		
	Detection of heteroresistant population or contamination		
Gradient test	Convenient and flexible	Relatively expensive	Quantitative
	Simple to perform	Relatively long incubation	
	Does not require expertise		
	Detection of resistance patterns		
Automated systems	Simple to perform	Relatively expensive	Semi-quantitative ***
Chromogenic media	Mass use and the possibility of automatisation	Not completely susceptible and specific	Qualitative with no interpretation criteria (S, I, R)
	Simple to perform	Time-consuming	
	Simple and fast interpretation	Limited spectra or single antibiotic	
		Relatively expensive	
		Screening only or required confirmatory identification	
		No MIC value	
MALDI-TOF MS	Rapid turnaround time	High cost of the MALDI-TOF MS	
	Simple to perform	Need further optimisation for each species and antibiotic combination	
	Low sample volume requirements	No MIC value	
	Low per-sample costs		
Genetic methods	Rapid	Limited spectra	Qualitative
	Highly accurate	Limited throughput	Semi-quantitative
	Sensitive	High cost	
	Reproducible		
	Increased ability to detect slow-growing or non-cultivable organisms		
Genomic methods	Highly accurate	High cost	Qualitative
	Sensitive	Time-consuming	
	Increased ability to detect slow-growing or non-cultivable organisms	Challenging interpretation of results	

* Qualitative; results are expressed as susceptible (S), susceptible, increased exposure (I), or resistant (R) based on established criteria from EUCAST. ** Quantitative; results are expressed as minimal inhibitory concentration (MIC) for each drug. Susceptibility reports should include interpretation of MIC, such as S, I, or R. *** Semi-quantitative; results are expressed as MIC using three to four antimicrobial dilutions for each drug. Precise MIC values cannot be established if the MIC falls below or above the three to four dilutions used in the test panel. Susceptibility reports include interpretation of breakpoint MIC as S, I, or R. MALDI-TOF MS—matrix-assisted laser desorption/ionization time-of-flight mass spectrometry.

**Table 2 antibiotics-11-00427-t002:** Troubleshooting algorithm for antimicrobial susceptibility testing quality control.

Factor	Influence	Suggested Solutions
Media (depth of agar)	Thin media yield excessively large inhibition zones and vice versa.	Measure agar depth carefully.
Composition of medium	Affects rate of growth of organisms; affects activity and diffusion of antibiotics.	Follow guidelines for an appropriate choice of media; perform quality control.
Antibiotic disks (potency)	Deterioration in content leads to smaller inhibition zone sizes.	Use a new lot of disks or unopened cartridge. Maintain majority of disk stock at −20 °C, only keep maximum of 1 week supply at 4 °C (be cautious of β-lactams, clavulanic acid-containing disks and imipenem).
Antibiotic disks—spacing	Disks too close together will cause overlapping zones. A smaller plate accommodates fewer disks	Place fewer disks on a plate (especially with very susceptible organisms)
Timing of antibiotic disk application	If placed long after swabbing plates, small zones of inhibition may form.	Apply disks within 15 min.
Reference strains for QC	Incorrect reference strain used for specific AST will lead to incorrect zone diameters—false alarm.	Follow guidelines for an appropriate choice of QC strains; perform quality control.
Inoculum density	Larger zones of inhibition with a light inoculum and vice versa.	Use McFarland standard or calibrator to carefully measure inoculum density and perform colony counts.
Incubation time	In most cases, ideal 16–18 h; less time than recommended gives unreliable results.	Follow guidelines for appropriate incubation time.
Temperature	If <35 °C larger zones of inhibition are seen and MRSA may go undetected.	Follow guidelines for appropriate incubation temperature.

QC—quality control.

## Data Availability

Not applicable.

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
