# Peer review of "Antimicrobial Susceptibility Testing: A Comprehensive Review of Currently Used Methods"

_antibiotics, 2022, doi:10.3390/antibiotics11040427_

Round 1

Reviewer 1 Report

Major Comments.

Reviews on the subject have been published, what is the knowledge gap that this particular review intends to fill?

What is novel in this review and what is the rational for putting up this review?

Has there been some recent knowledge that needs to be harmonized and disseminated?

Minor comments

Line 52: Which cases are been referred to .? Please clarify 

Line 71: "..... local and clinical laboratories..." Clarify local i.e. the difference between these 2

Line 75: Capitalize the first letters in Center for Disease Control and Prevention

Line 83-84: Sentence is ambiguous, clarify

Line 95-95: Modify to read, In a retrospective study by Puzniak et al.....

Line 129: What is the difference between this title and Line 58

Line 134: Do you mean disc diffusion?

Section 3.1.1: I will suggest the prospect and disadvantage of each method be reviewed too 

Line 156: Which method is the BDM??? Provide a brief on it.

Line 179: ".. advantages properties...." correct grammar 

Line 179-180: "It allows for determination of MIC while remaining simple for use" Sentence is ambiguous please clarify

Line 208: Use "+" instead of "with" i.e. Cefotaxime+clavulanic acid

Line 245: modify to ".... ease of performance and applicability"

Line 317: Section 2 here and also section 2 on line 58???

Lines 401-409: Provide examples of such techniques

Line 441-467: Similarly provide examples e.g. Alere genotype kits

Line 486: I suggest the heading be Mass Spectrometry

Table 1: "Good four facets tool"???

Table 1: Can you classify Chromogenic agar as automated???

Line 598: Delete the phrase "..when the''"

Reviewer 2 Report

The manuscript has presented a review of methods used for assessment of antimicrobial resistance, showing aventages and disadventages of these techniques. The review is up-to-date; well written and supported with distinct figures and tables. This is a good compendium of antimicrobial susceptibility testing for microbiologists and cliniciants. Some minor corrections are needed in the text; "clinical" (l. 108), "assays" for "easy" (l. 164), "eight" for "8" (l. 211), "third" for "3rd" (l. 758).

Reviewer 3 Report

Rating the Manuscript

  • Originality/Novelty: the question is not original but it is well defined.
  • Quality of Presentation: the article is written in an appropriate way and the data is presented appropriately.
  • Interest to the Readers: I believe the conclusions are of interest of the readership of the Journal. In my opinion, the paper could attract a wide readership within the clinical and research field working in antimicrobial resistance, especially at early career professionals since it touches the basics.
  • Overall Merit: the authors put together in a very nice way not only AST techniques, but also put in perspective why they need to be performed and quality control steps to follow to ensure reproducibility.
  • English Level: the English language is appropriate and understandable but since English is not my first language, I do not feel qualified to judge about language and style.

Brief summary

The authors reviewed the current antimicrobial susceptibility testing (AST) methods applied in clinical microbiology laboratories, as well as emerging AST techniques. Aside from presenting advantages and disadvantages of each technique, they elaborated on the rationale of performing ASTs and quality control of the process.

General concept comments

The manuscript is clear and relevant for the field but one could argue that to review AST methods is not new since they have been reviewed in previous peer-reviewed articles e.g. doi:10.1111/jam.14704 or doi: 10.3390/diagnostics9020049. Nevertheless, AST methods have thoroughly discussed and gaps in knowledge adequately addressed. Additionally, the inclusion of the rationale of performing ASTs and quality control adds some novelty to the work in my opinion.

The cited references are for the most part current and the manuscript does not include an abnormal number of self-citations. References seem to be formatted according to the journal guidelines.

The manuscript is not presented in a well-structured manner since the manuscript sections are messy:

  1. The emergence of antimicrobial resistance and overlooked pandemic
  2. The rationale for performing susceptibility testing
  3. The rationale for performing antimicrobial susceptibility testing (why do these two sections have a near-equal title?)

                3.1. Classical methods

                                3.1.1. Dilution methods: broth dilution and agar dilution

                                3.1.2. Antimicrobial gradient method

                                3.1.3. Disk diffusion test

                                3.1.4. Chromogenic agar media for detection of antimicrobial-resistant bacteria

                                3.1.5. Colourimetric tests for detection of antimicrobial-resistant bacteria

2 (number 2 again?). Current Technologies for Rapid AST

                2.1. Automated and semi-automated devices for antimicrobial susceptibility testing

                2.2. Molecular-based techniques for resistance detection

                2.3. Nucleic acid amplification-based methods

                                2.3.1. Polymerase chain reaction (I would argue that NAAT are molecular-based techniques,                                          consequently, they should be within point 4.2 and not as an independent point 4.3.)

                                2.3.2. DNA-microarrays

                                2.3.3. Whole-genome sequencing in antimicrobial susceptibility testing

                2.4. Matrix-assisted laser desorption/ionization time-of-flight mass spectrometry                                              (MALDI-TOF)

3 (Number 3 again?). Selection of antimicrobial drugs for susceptibility testing, interpretation, and reporting

  1. Quality Assurance in Antimicrobial Susceptibility Testing
  2. Near-future perspectives for antimicrobial susceptibility testing
  3. Conclusion

The statements and conclusions drawn are coherent and supported by the listed citations.

Figures 1 and 3 present icons/ elements from BioRender. I could not find any reference to BioRender within the manuscript text. Did the authors obtained the peer-review publication rights of BioRender-generated images through the purchase of a premium account? If not, images 1 and 3 cannot be used for publication purposes. Please take action accordingly.

Specific comments:

Line 35: specify which are the six-leading pathogens.

Line 38: COVID-19 also put on hold many AMR surveillance projects because funds were re-directed to fight the pandemic. Please comment on that.

Line 45: The World Bank… How does this statement relate to AMR?

Line 72: “…the most important being…” Most important based on what?

Line 87: I would add that also for secondary and tertiary hospital where more illness of chronic nature are treated and chances are the patients there already went through more antimicrobial courses compare to primary hospitals, increasing in this way antimicrobial selective pressure.

Line 89: Please describe what these different antibiogram types refer to.

Line 95:change “in the…” for “In a…”

Line 100: change :conclude” for “concluded”.

Line 108: “…which 48h.” Is this for all bacterial species or there slow-growing fastidious bacteria that may take longer? Does this time consider the time required to obtain a pure bacterial culture since the collection of the specimen?

Line 118: change “a rapid and reliable test” for “more rapid and reliable tests”.

Line 135: “…representing the reference methods.” Missing a reference to support such claim.

Line 136: Why obtaining the MICs is advantageous? I think this point needs further clarifications for readers.

Line 142: Uses serial two-fold dilutions.

Line 144: as described where?

Line 158: two-fold.

Line 162: is a blue color indicator?

Line 168: serial dilutions.

Line 179: advantageous instead of advantages.

Line 181: re-phrase “Most correlation studies…”. What category do you refer to? SIR categories?

Line 184: Etest needs to be defined the first time is mentioned in the text because it is not the strips brand of BioMerieux.

Line 185: a correlation can be positive or negative, please re-phrase this part since the point is not clear.

Line 209-10: “The strip..>” phrase doesn’t read well.

Line 235: cause disease in humans?

Line 240: change “were” for “are”

Line 246: “…and antibiotics.” Missing a reference here.

Line 247-8: change “isolated bacteria and the type of sample from which the isolate is obtained” for “the bacterial species and the type of sample the isolate was obtained from.”

Line 250: change “detect” for “determine”.

Line 255: automatic systems?

Line 263: the identification of what?

Lines 263-4: shortening the time by how long?

Line 269: “…is more reliable than MIC…” How so?

Line 274: change “indicates” for “suggests”.

Line 291: change resistance for resistanT.

Line 306: missing a reference.

Line 308: colourimetric tests section needs further information about how this test look like, how bacteria are inoculated etc. like for previous sections.

Line 318: Automated and… section: are all these automated devices based on micro-dilution susceptibility testing? If so, please indicate so.

Line 333: one card per isolate?

Line 345 and 347: what's the capacity of the system per run in terms of bacterial isolates?

Line 355: Conventional here means...?

Line 359: the "s" from "system" in the previous point was upper case, this one doesn't. Please unify nomenclature.

Line 369:  substitute “were not refrigerated” for “do not need refrigeration.”

Line 379: substitute “the actual standard” for “current standards”.

Line 392: “Another standard…” Another? Where is described the previous one?

Line 395: italic for Streptococcus?

Line 404: how much faster?

Line 419: “…increases.” Missing a reference for that statement.

Line 427: how does it reduce risk of sample contamination? I would argue that because higher sensitivity, the chances of sample (cross) contamination are higher.

Line 428: which systems?

Line 458: incomparable?

Line 628: change “allow” for “help”.

Lines 656-7: change “with the propensity to induce multidrug resistance” for “that pose higher AMR selective pressure.”

Line 664: change “prefer” for “choose”.

Line 701: is the troubleshooting algorithm based on a published document? If so, please cite.

Table 2, 3rd row, last column: add a period after “cartridge” and every sentence, it is confusing that in this specific case for example, the text continues with “Maintain majority…” starting with an upper case letter when there wasn’t a period before. Also because that sentence ends by a period (the only one).

Table 2, 5th row: timing of application of what?

Table 2, 5th row, 2nd column: small zones of what?

Table 2, 6th row: also reference strains kept for long periods of time even at -80C may show different AST results and new ones must be purchased.

Table 2, 8th row, 2nd column: change “less” for “shorter”; also, what about longer incubation times?

Table 2, 9th row, 2nd column: larger zones of what? Tables should stand by themselves.

Lines 717-8: change “using the software support” for “the use of the automated system software supports”.

Line 731: change “used” for “considered”.

Line 736: doesn't read well, please re-phrase.

Line 739: chance “on” for “against”.

Line 741: this new paragraph should follow the previous one and not be an independent one.

Line 760: I don't think flow cytometry has been mentioned before in the manuscript at all. So this paragraph comes out the blue. In my opinion, it needs to be elaborated a bit more. Additionally, it lacks references for the statements presented. Same for the technique discussed in the next paragraph i.e. isothermal microcalorimetry.

Line 781: “The molecular-based…” eliminate the “the”.

Line 785: acquisition of what? The machine?

References 41, 50, 51, 73, 75, 82, 83, 85, 86, 88, 89, 92, 93, 96, 109, 110, 111, 116, 118, 121, 122, 124, 128, 129 (missing a period), 132, 141, 145, and 146: eliminate the upper case from the first letter of every word that doesn’t need it.

Reviewer 4 Report

This is an excellent review focusing on the various antimicrobial susceptibility testing (AST) methods, which are presently used or possibly applicable in the conceivable future, as well as their advantages and disadvantages. I especially appreciate the comprehensive nature and clear description of each susceptibility testing method with examples. The manuscript is very well written; the pictorial representation of methods and workflow is also admirable. I have no major suggestions, only minor points for the authors to consider.

  1. The author should check the numbering system of the topic once again and correct them accordingly.
  2. In my opinion, the author can consider briefly including the different antimicrobial resistance mechanisms employed by bacteria (PMID: 35264857, PMID: 31294229) in the introductory section to increase the scope to reach a broader audience. This is important especially as they have mentioned various genes in the nucleic acid amplification method and the reader could be able to correlate more if introduced before in the manuscript.
  3. Another thought from a reader's point of view, the author could include their thoughts as to future prospective about strategies that could be employed to reverse back the antimicrobial resistance to make bacteria susceptible based on the literature evidence. A few examples to be included are as follows: (essential target in the bacterial life cycle: Cell envelope biogenesis factors- outer membrane lipid transport, peptidoglycan cell wall (PMID: 32908144, PMID: 35274942, PMID: 33917043), essential cell division proteins (PMID: 24755375) (targeting non-essential targets that are critical during processes of host colonization and infection. For example virulence-conferring factors (PMID: 30410623), Efflux pump inhibitors ( PMID: 32665275), targeting secondary resistome (PMID: 28198411, PMID: 24277026), inhibitors of energy metabolism (PMID: 29473841), epigenetic modulators (PMID: 31740560).

Apart from these, authors should be congratulated for their commendable effort in putting this review.
